# Structural Consequences of Deproteinating the 50S Ribosome

**DOI:** 10.3390/biom12111605

**Published:** 2022-10-31

**Authors:** Daniel S. D. Larsson, Sandesh Kanchugal P, Maria Selmer

**Affiliations:** Department of Cell and Molecular Biology, Uppsala University, SE 751 24 Uppsala, Sweden

**Keywords:** ribosome assembly, RNA structure, RNA folding, RNA-protein interactions, single-particle cryo-EM

## Abstract

Ribosomes are complex ribonucleoprotein particles. Purified 50S ribosomes subjected to high-salt wash, removing a subset of ribosomal proteins (r-proteins), were shown as competent for in vitro assembly into functional 50S subunits. Here, we used cryo-EM to determine the structures of such LiCl core particles derived from *E. coli* 50S subunits. A wide range of complexes with large variations in the extent of the ordered 23S rRNA and the occupancy of r-proteins were resolved to between 2.8 Å and 9 Å resolution. Many of these particles showed high similarity to in vivo and in vitro assembly intermediates, supporting the inherent stability or metastability of these states. Similar to states in early ribosome assembly, the main class showed an ordered density for the particle base around the exit tunnel, with domain V and the 3′-half of domain IV disordered. In addition, smaller core particles were discovered, where either domain II or IV was unfolded. Our data support a multi-pathway in vitro disassembly process, similar but reverse to assembly. Dependencies between complex tertiary RNA structures and RNA-protein interactions were observed, where protein extensions dissociated before the globular domains. We observed the formation of a non-native RNA structure upon protein dissociation, demonstrating that r-proteins stabilize native RNA structures and prevent non-native interactions also after folding.

## 1. Introduction

The ribosome is a large macromolecular complex translating mRNA into protein in all domains of life. Structural biology has elucidated the architecture of ribosomes, where each subunit consists of structured ribosomal RNA (rRNA) stabilized by ribosomal proteins (r-proteins) that use positively charged loops and tails linked to globular domains to stabilize the tertiary structure of the rRNA. The intricate assembly of ribosomes is, in all organisms, a tightly regulated process that has been fine-tuned by evolution. 

The assembly of the 50S subunit from *E. coli* has been extensively studied in vitro, and methods were early developed to reconstitute active subunits from components [1]. Another in vitro approach is to promote the dissociation of r-proteins through high-salt wash of ribosomal particles. Proteins were shown to dissociate from the core at defined concentrations of, e.g., LiCl [2] in a non-denatured state [3,4]. The observed order of leaving r-proteins nearly agrees with the reverse of the in vivo assembly order followed by quantitative mass spectrometry [5] as well as the in vitro reconstitution protocol [6]. A subset of the ribosomal proteins that dissociate during salt wash also exchanges in vivo, presumably as a mechanism by which damaged proteins can be continuously replaced [7].

In vivo, *E. coli* makes use of a multitude of auxiliary RNA modification enzymes and ribosome assembly factors to fold and assemble ribosomes from three rRNAs and 54 r-proteins (reviewed in [8,9]). Despite the complexity of the process, the assembly time for a mature ribosome in vivo in exponentially dividing *E. coli* has been estimated to only 2 min [10]. Our structural understanding of the folding and assembly of ribosomes has greatly benefitted from progress in cryogenic electron microscopy (cryo-EM), where classification schemes in single-particle reconstruction have proven essential to studying these heterogeneous particle ensembles. To enrich ribosome assembly intermediates in vivo for these studies, bacteria have been perturbed by assembly inhibitors [11,12], the knock-down of r-proteins or assembly factors [13,14,15], or subjected to pull-downs with assembly factors as bait [16,17].

In one such study, a bL17 depletion strain was used to generate ribosomal 50S sub-particles [13], allowing the identification of several cooperative folding blocks. In agreement with previous studies [12], a model of three major routes of LSU folding upon bL17 perturbation was proposed, where the central protuberance (CP), the particle base, and the L7/L12 stalk formed in different orders. 

Structural studies of assembly intermediates from LSU in vitro reconstitution [18] showed five distinct precursors that shared many features with in vivo assembly intermediates from the bL17-depleted strain. The later intermediates displayed an intricate variety of folding states of the protuberances and the peptidyl transfer center (PTC).

Salt-washed LiCl core particles have been demonstrated to be assembly-competent, allowing reconstitution into active 50S subunits [1,6]. They also work as in vitro substrates for RlmF, an early rRNA modifying enzyme that lacks activity on naked RNA [19], supporting at least local similarity to states occurring during ribosome assembly in vivo. To date, there is no high-resolution structure of these particles, but early negative-stain electron microscopy showed a high degree of heterogeneity where approximately 40% of the ribosomes remained as compact particles lacking the three characteristic protuberances of the LSU [20]. 

Here, we set out to further characterize the structures of LiCl core particles using single-particle cryo-EM, with the goal to characterize the high-resolution structure and composition of these particles. We identified six major types of 50S sub-particles with a wide range of sizes and a multitude of variants of each of these. The method allowed observation of core particles smaller than the so-far identified 50S assembly intermediates and the opportunity to examine the effects of removing particular r-proteins. A comparison of these particles enabled us to identify stability dependencies between rRNA helices and r-proteins, reconstruct possible disassembly and assembly pathways, and demonstrate strong similarities between intermediates formed during disassembly and assembly.

## 2. Materials and Methods

### 2.1. Preparation of LiCl Core Particles

Frozen cell pellets of *E. coli* ∆*ybiN* from the Keio collection (JW5107, [21]) (24 g, wet weight) were resuspended in 100 mL of opening buffer (20 mM Tris-HCl pH 7.5, 100 mM NH_4_Cl, 10 mM Mg(OAc)_2_, 0.5 mM EDTA and 3 mM β-mercaptoethanol) containing 5 µg/mL DNase I (Sigma-Aldrich, St. Louis, MI, USA) and cOmplete™ EDTA-free protease inhibitor cocktail (Roche, Basel, Switzerland) and the cells were opened using a CF Cell Disrupter (Constant Systems Ltd., Daventry, UK). The lysate was cleared by centrifugation (Sorvall SS-34 at 16,000 RPM for 45 min) and filtration (0.45 µm pore size). Crude ribosomes were obtained by pelleting (Beckmann Type 45Ti at 40,000 RPM for 18 h) 25 mL fractions of lysate through 27 mL cushions of sucrose buffer (20 mM Tris-HCl pH 7.5, 500 mM NH_4_Cl, 10.5 mM Mg(OAc)_2_, 0.5 mM EDTA-Tris, 1.1 M sucrose, 3 mM β-mercaptoethanol). The ribosome pellets were dissolved in wash buffer (20 mM Tris-HCl pH 7.5, 500 mM NH_4_Cl, 10.5 mM Mg(OAc)_2_, 0.5 mM EDTA-Tris, 7 mM β-mercaptoethanol), and sedimented through sucrose cushions a second time (25 mL of ribosomes and 1.5 mL of sucrose buffer in a Beckmann SW32Ti at 24,800 RPM for 16 h). The crude ribosomes were separated on pre-formed linear 15–30% sucrose gradients (Beckmann SW32Ti at 19,300 RPM for 17 h) in TC buffer (20 mM Tris-HCl pH 7.5, 60 mM NH_4_Cl, 5.25 mM Mg(OAc)_2_, 0.25 mM EDTA-Tris, 3 mM β-mercaptoethanol), and the 70S peak was pooled and pelleted (Beckmann Type 45Ti at 45,000 RPM for 18 h). To obtain individual subunits, 70S pellets were resuspended in SU buffer (TC buffer but with 3 mM Mg(OAc)_2_) and subunits separated on pre-formed linear 15–30% sucrose gradients in SU buffer (Beckmann SW32Ti at 19,300 RPM for 16 h). The 50S subunit fractions were pooled and pelleted (Beckmann Type 45Ti at 45,000 RPM for 18 h). The pellets were resuspended in LiCl-wash buffer (10 mM Hepes-KOH pH 7.5, 10 mM Mg(OAc)_2_, 3.5 M LiCl) and incubated for 5 h on ice. The 50S LiCl core particles were pelleted (Beckmann Type 90Ti at 44,000 RPM for 4 h), resuspended in SU buffer, and separated on pre-formed linear 15–30% sucrose gradients in SU buffer (Beckmann SW32Ti at 19,900 RPM for 16 h). The 50S LiCl core particle fractions were pooled, pelleted (Beckmann Type 90Ti at 45,000 RPM for 17 h), and resuspended in storage buffer (20 mM Tris-HCl pH 7.5, 50 mM NH_4_Cl, 5 mM Mg(OAc)_2_, 3 mM β-mercaptoethanol). The concentration of 50S LiCl core particles was measured by 260 nm absorbance (assuming 1 A_260_ unit equals 38 nM) and aliquots were flash-frozen and stored at −80 °C.

### 2.2. In Vitro Methylation Assay

Assays were performed in duplicates. Each reaction contained 30 pmol 50S-LiCl particles in reaction buffer (50 mM HEPES-KOH pH 7.5, 20 mM NH_4_Cl, 100 mM KCl, 2.5 MgOAc, 8 mM β-mercaptoethanol). The addition of 400 pmol unlabeled S-Adenosyl-L-methionine (Sigma-Aldrich, USA) doped with 16 pmol S-[methyl-^3^H]-adenosyl-L-methionine (4 Ci/mmol; PerkinElmer, Waltham, MA, USA) was followed by the addition of 40 pmol RlmF, or buffer (negative control) to a final volume of 30 μL. All reactions were incubated at 37 °C for up to 30 min, quenched in 2 mL of ice-cold 10% trichloroacetic acid, and incubated on ice for 10 min. The precipitated reactions were applied to BA85 nitrocellulose filters (Whatman, Little Chalfont, UK) under vacuum and washed five times with 7 mL of cold 10% trichloroacetic acid. The washed filters were placed in vials containing 5 mL of Filter Safe scintillation cocktail (Schleicher & Schuell GmbH, Keene, NH, USA), shaken for 30 min, and counted in an LC6500 scintillation counter (Beckman, Brea, CA, USA). For calculation of the fraction of modified particles, the CPM per pmol of doped SAM mix was measured. This allowed conversion of CPM per pmol of ribosomal particles to pmol of modified particles.

### 2.3. Cryo-EM Sample Preparation

A 300-µL reaction containing 5.2 nmol of C-terminally His-tagged RlmF and 1.2 nmol of 50S LiCl core particles in imaging buffer (50 mM Hepes-KOH pH 7.5, 100 mM NH_4_Cl, 10 mM Mg(OAc)_2_, 0.5 mM EDTA-Tris, 6 mM β-mercaptoethanol, 0.5 mM sinefungin (Sigma-Aldrich, USA), 0.5 U/mL of RiboLock RNase inhibitor (Thermo Fisher Scientific, Waltham, MA, USA)) was incubated with 200 μL of Ni-Sepharose (GE Healthcare, Uppsala, Sweden) for 30 min at 25 °C under gentle agitation in a spin-filter and, subsequently, centrifuged at 0.5 RFC for 3 min. The Ni-Sepharose beads were washed twice by incubation for 10 min in 500 µL of imaging buffer with 0.2 mM sinefungin, 0.1 U/mL of RiboLock RNase inhibitor and 30 mM imidazole, and the ribosome-RlmF complex was eluted in 200 uL of imaging buffer with 0.2 mM sinefungin, 0.1 U/mL of RiboLock RNase inhibitor and 300 mM imidazole. The eluate was sedimented in 50-μL aliquots through 50 µL of 50% sucrose cushion in imaging buffer (Sorvall RC M150GX for 90 min at 80,000 RPM at 4 °C). The pellets were resuspended in 10 μL of imaging buffer and stored on ice before being flash-frozen on cryo-EM grids. The protein content was verified by SDS-PAGE and LC-MS/MS analysis of the lane cut from the SDS-PAGE gel. Sample preparation is summarized in Appendix A.

For the LC-MS/MS analysis, the sample was digested by trypsin and the peptides were subjected to reverse phase separation on a C18-column with a 90-min gradient and electrosprayed to a Q Exactive Orbitrap mass spectrometer (Thermo Finnigan, San José, CA, USA). Tandem mass spectrometry was performed by high-energy collision dissociation fragmentation. The data were analyzed using Proteome Discoverer 1.4 (Thermo Fisher Scientific) using the Sequest algorithm against a database of the *E. coli* K-12 BW25113 proteome (the parental strain to the Keio collection). The search criteria for protein identification were set to at least two matching peptides of 95% confidence level per protein.

### 2.4. Cryo-EM Grid Freezing and Data Collection

The cryo-EM specimens were prepared on QuantiFoil R 2/2 grids with 2-nm continuous carbon (QuantiFoil Micro Tools GmbH, Großlöbichau, Germany). The grids were glow-discharged for 40 s at 20 mA and 0.38 mBar using an EasiGlow (Ted Pella, Inc., Redding, CA, USA). Plunge-freezing was done using a Vitrobot mark IV (Thermo Fisher Scientific, USA) using 3 µL of sample at 160 nM ribosome concentration after 30 s of incubation on the grid at 100% humidity and 10 °C and blotting for 3.5 s. Data were collected on two different occasions on the same Titan Krios microscope from two different grids from the same batch. The microscope was equipped with a Falcon-III direct electron detector used in integrating mode. The nominal magnification was 75,000×, corresponding to a pixel size of 1.09 Å in the specimen plane, and the acceleration tension was 300 kV. On the first occasion, movies were collected with 66.2 e^−^/pixel/second during 0.773 s for a total dose of 41.0 e^−^/Å^2^, and the sample stage was set to either 0° (3052 movies) or 15° (2196 movies) tilt. In the second session, movies were collected with 68.4 e^−^/pixel/second during 0.770 s for a total dose of 42.2 e^−^/Å^2^, and the sample stage was set to a 30° tilt (6120 movies). Data collection parameters are summarized in Appendix A.

### 2.5. Cryo-EM Data Processing

The 3D single-particle reconstruction was done using RELION version 3.0 [22] following the conventional RELION workflow [23]. The 0°, 15°, and 30° tilt micrographs were initially processed separately and later combined. Movies were motion corrected in 5 × 5 patches using the RELION implementation of the MotionCor2 algorithm [24]. Per-micrograph defocus parameters were estimated using gCTF version 1.18 [25]. Particle picking was done using the Laplacian-of-Gaussian algorithm in RELION. Local CTF parameters were estimated for the tilted micrographs using gCTF. An ab initio 3D initial model was generated from primary data to prevent model bias. Multiple rounds of 2D and 3D classifications were performed to prune the data down to a set of 384,374 particles. Automatic 3D refinement, per-particle refinement of CTF parameters, and particle polishing were performed several times until no further improvement in resolution could be achieved. RELION version 3.1 [26] was then used to estimate higher order aberrations and anisotropic magnification, followed by further CTF-refinement and particle polishing, to produce the final 2.84-Å consensus reconstruction. The mask for estimating resolution was generated by masking the final reconstruction with a 155-Å spherical mask to remove artifacts at the rim of the reconstruction followed by low-pass filtering to 15 Å, thresholding that map at a sufficiently low value to encompass diffuse density associated with the particle map but excluding solvent noise (0.012), expanding the mask first by 5 pixels, and finally adding a 5-pixel soft edge. Estimation of local resolution and local low-pass filtering was performed using the RELION algorithm using a B-factor determined by fitting to an approximately linear regime at a high resolution of the Guinier plot (−32.2 Å^2^). The efficiency of the orientation distribution (E_od_) parameter was calculated using cryoEF [27] and the distribution of Euler angles was plotted (Appendix A). All but one class had E_od_ values in the range of 0.6–0.8 (Appendix A), which can be expected for ribosome data sets [27]. To identify distinct particle subsets, 3D classification without alignment was performed based on the consensus reconstruction. The particles in the consensus reconstruction were classified into six classes, chosen based on initial testing with more classes, and each class was refined separately according to the above procedure. Subsequently, classes were further 3D-classified in the same manner in tiers, each time requesting four classes. Data processing is summarized in Appendix A.

### 2.6. Model Refinement in Cryo-EM Maps

A high-resolution crystal structure of the *E. coli* ribosome (PDB ID 4YBB [28]) was rigid-body fitted into the consensus map using UCSF Chimera version 1.14 [29]. The pixel size was refined to 1.07 Å by monitoring the map-model fit while varying the pixel size. Large disordered regions were pruned from the model and refinement was performed using real_space_refine in Phenix version 1.19 [30] with reference model restraints to PDB ID 7K00 [31] and interactive rebuilding using Coot version 0.9 [32]. In the final deposited model, all nucleotides not supported by the density were removed. No water oxygens or ions were modeled. Figures were prepared using ChimeraX version 1.4 [33]. Model refinement parameters are summarized in Appendix A.

### 2.7. Occupancy of Structural Elements and Hierarchical Clustering

The atomic model of a mature ribosome (PDB ID 4YBB), as well as a model refined against the consensus reconstruction, were rigid body fitted to each class reconstruction using UCSF Chimera 1.14 [29], and the map values were calculated at the position of each heavy atom (i.e. non-hydrogen) in the LSU using the phenix.map_value_at_point program. For these calculations, unsharpened maps produced by Relion 3D refinement were used, which are on an absolute gray scale. Maps were low-pass filtered to 5 Å to avoid spurious results due to local misalignments between the maps and the models, and the pixel size was adjusted to the calibrated pixel size.

The occupancy was calculated for each structural element (chain or 23S rRNA helix) as the fraction of atoms with a map value above a threshold of 0.05. The threshold was confirmed by manual inspection of the maps. The values were normalized according to [13], i.e., by dividing by the value by the maximum observed value for a particular feature across all classes or by the median values across all features and maps, whatever was larger.

Manual inspection to gauge occupancies of r-protein was performed by identifying the map threshold where the strongest recognizable feature could be seen. Density for 23S rRNA helices was classified as being near-native, slightly distorted, highly distorted, or disordered, which were translated into numerical values 0.1, 0.085, 0.045, and 0, respectively, for clustering and plotting purposes.

The boundaries for the 23S rRNA helices were taken from [34], specifically the segmentation in the *E. coli* secondary structure map downloaded from the Ribosome Gallery on 22 September 2020 (http://apollo.chemistry.gatech.edu/RibosomeGallery/bacteria/E%20coli/LSU/E_coli_LSU_Helices_2.png), which differs slightly from the original publication. RNA helices not named in the map were given the number of the helix immediately preceding with an additional letter or the next number if available, cf. Appendix A.

Hierarchical clustering using the complete linkage method and the Euclidean metric was calculated using the function scipy.cluster.hierarchy.linkage in the SciPy library (https://www.scipy.org/, accessed on 10 August 2020), both for classes (columns) as well as features (rows). R-proteins that could not be identified in any of the classes were excluded from the analysis. The clustering of features was rather sensitive to the chosen parameters (e.g., reference model, threshold, low-pass filtering, occupancy normalization, exclusion/inclusion of r-proteins, etc.), but the general trends were reproducible.

## 3. Results

### 3.1. Sample Preparation

LiCl core particles were prepared from active 50S subunits of *E. coli* ∆*ybiN*, a strain lacking the RlmF-mediated methylation of A1618, by salt wash and sucrose gradient purification (Appendix A). In an in vitro tritium-labeling assay, these LiCl core particles could be methylated by RlmF [19], and methylation showed saturation after 10 min at ~20% modification (Appendix A). With the aim to determine the structure of RlmF bound to a pre-ribosomal-like substrate, the sample was enriched for such particles by pull-down with His-tagged RlmF in the presence of the SAM analog sinefungin (Appendix A). The pull-down yield was 4-fold higher in the presence of RlmF and sinefungin compared to the negative control without both. In the resulting population, RlmF was by SDS-PAGE judged to be present at approximately the same stoichiometry as uL2 (Appendix A) and LC-MS/MS analysis showed a similar or higher signal as for the stably bound r-proteins. However, no distinct density for RlmF could be found in any of the particle classes after single-particle reconstruction, indicating that it dissociated after the pull-down. 

### 3.2. High-Resolution Cryo-EM Reconstruction of the 50S LiCl Core Particle

Cryo-EM imaging, single-particle reconstruction, and a multi-tiered classification scheme were used to determine the structure of the LiCl core particle (Figure 1). Because of strong preferred orientation [20], data collected at 15° and 30° tilts were used to improve the angular sampling (Appendix A). The micrographs showed heterogeneous particles (Appendix A) and several of the 2D class averages had one disordered side (Appendix A). 

### 3.3. LiCl Washing of the LSU Produces a Range of Sub-Particles

A consensus reconstruction from ~384,000 particle images produced a map at 2.84 Å resolution (Figure 2 and Appendix A). The reconstructed density is dome-like with the ordered part at the 50S solvent side, centered on the expanded nascent-chain exit tunnel (Figure 3a,b). None of the three characteristic protuberances of the LSU are ordered.

Classification resulted in six major classes, numbered 1–6 according to size (Figure 1 and Appendix A). Three levels of further classification generated several sub-classes, where the largest sub-class 5-5 collected approximately 1/3 of the initial particles. Model refinement (Figure 3a and Appendix A) and some of the analysis and figures are based on the consensus map, which is very similar to class 5-5; importantly, all conclusions are also valid for this particle.

Most classes could be resolved to better than 4.5 Å resolution, but the smallest and largest particles could only be resolved to approximately 9 Å resolution (Appendix A), probably due to a low number of particles and, for the smallest particles, heterogeneity in structure as well as the presence of r-proteins. All particles have resolutions that allow interpretation on the level of presence or absence of double-helical RNA elements.

### 3.4. The LSU Is the Most Stable at the Solvent-Side

In the majority of the particles, more than half of the 23S rRNA is folded, including domains 0, I, III, VI, most of domain II, and the 5′ half of domain IV (i.e., H62–H67, from now on referred to as sub-domain IV_5′_) (Figure 2b). The folded regions are at the solvent side and particle base of the LSU and to a high degree correlate with the presence of r-proteins (Appendix A). Several rRNA helices appear as folded but flexible low-resolution features. The three major protuberances and the subunit interface, including functionally important regions, such as the PTC and binding sites for tRNAs and GTPases, are unstructured (Figure 2). In addition, many single nucleotides involved in native tertiary contacts are disordered (Figure 2b).

### 3.5. The LiCl Core Particle Classes Are Analogous to In Vivo and In Vitro Assembly Intermediates

Classes 2, 5, and 6 show strong similarities to in vitro reconstitution intermediates [18] and to in vivo assembly intermediates under bL17 depletion [13] (Appendix A), although they all include bL17. Major class 1 is smaller than any previously studied 50S assembly intermediate and seems to represent a minimal stable core of the LSU.

To arrange the particles into a tree that allowed identification of cooperativity between binding of r-proteins and folding of rRNA, automated (Appendix A) and manual (Appendix A) occupancy estimations were subjected to hierarchical clustering (Appendix A). The classes grouped into smaller (classes 1, 2, 3, and 5-1), medium-sized (classes 4 and 5, except 5-1 and 5-8), and larger particles (classes 5-8 and 6) (Appendix A), analogous to early, intermediate, and late 50S assembly intermediates (Appendix A). In the small particles, most of 23S rRNA domains 0, I, III, VI, and a core of domain II are folded (Figure 4). In the medium-sized particles, more of domain II, as well as sub-domain IV_5′_ and parts of domain V are folded. The large particles are mature-like, but parts of sub-domain IV_3′_ and parts of domain V are unfolded. The clustering of structural features (rows in Appendix A) approximately agrees with previously identified folding blocks [13].

### 3.6. The Misfolded 3′ Strand of Helix H73

Non-native density at the base of the L7/L12 stalk shows up in several classes (1-1, 2-1, 3-1, 4, and 5-2), as a low-resolution protruding loop at low map thresholds (Figure 5 and Appendix A). One arm of the loop consists of H97 (Figure 5d), slightly shifted compared to the native particle. The other arm, with the appearance of an rRNA stem-loop, emerges outside the particle between H1, H94, and H97. Intriguingly, H73 is natively positioned just inside in most “non-star” particles, but absent in all of the “star” classes (Appendix A). Helix H73 is at a 4-way junction, but the region of its 3′ strand (Figure 5b) can, according to secondary structure prediction, form an alternative stem-loop (Figure 5c). A predicted 3D model of the stem-loop is compatible in size with the observed non-native density (Figure 5d). Closing the loop, an unidentified protein seems to bind to the top of helices H97 and the misfolded H73. Possibly, this could be uL6, which natively binds H97 and is detected at low levels in the LC-MS/MS data (Appendix A).

### 3.7. R-Proteins Leave the 50S According to In Vivo Assembly Groups

A number of proteins appear to be present across all classes (uL3, uL4, bL17, bL20, bL21, uL22, uL23, uL29, and bL34), and in most cases also uL13 and uL24 (Appendix A, Appendix A). All of these, except bL32, have firm support in LC-MS/MS data (Appendix A). These proteins are all early binders in vivo, as identified by pulse-labeling coupled with MS analysis [5] (Appendix A). Furthermore, four proteins that are only found in larger particles (uL2, uL14, bL19, and bL32) bind in the subsequent step in vivo. 

Fifteen proteins do not have clear density in any of the classes (uL1, uL6, bL9, uL10, uL11, bL12, uL15, uL16, bL25, bL27, bL28, bL31, bL33, bL35, or bL36), although diffuse density prevented unambiguous assessment of some proteins. Trace levels of uL1, uL5, uL6, bL9, uL11, uL15, and bL31 were detected by LC-MS/MS (Appendix A), suggesting that some of these r-proteins remain bound to disordered rRNA.

The high solubility of the CP-associated r-proteins uL5, uL18, and bL25 destabilizes the CP in the LiCl-washed ribosomes. The CP and the 5S rRNA are only discernable in the largest particle classes, 6 and 5-8 (Appendix A), together with uL5 and uL18 at low occupancy in class 6, and uL18 in class 5-8. 

For the three largest r-proteins in the consensus reconstruction (uL2, uL3, and uL4), the long loops and tails that in the native particle interact with the 23S rRNA are invisible (Appendix A). The interacting rRNA helices are in some cases unstructured and in other cases folded but in a non-native position.

### 3.8. Comparison of Particle Classes Show Dependencies between Different Structural Elements 

Based on the clustering analysis, structural consequences of the removal of certain r-proteins and of unfolding of rRNA were analyzed by pair-wise comparison between the particle classes. This revealed folding and stability dependencies that are mostly additive (Figure 6).

#### 3.8.1. Sub-Domain III_tail_ Can Stabilize Sub-Domain IV_5′_ in Absence of uL2

Class 1-1, the smallest reconstructed particle, only consists of domains I (except H25), III, VI, 0 (H26, H26a, and H61) and IV_5′_ (particularly strong density for H63) (Figure 4). This is remarkable since the folding of sub-domain IV_5′_ in other classes coincides with the presence of uL2 (see below). Class 1-2 has additional density for the core of domain II, a more mature-like domain 0 and H25 of domain I, but less density for domain IV and the distal half of domain III (helices H54–59, hereafter called sub-domain III_tail_ [36]).

Based on these structures, we hypothesize that in lieu of uL2, sub-domain III_tail_ can stabilize sub-domain IV_5′_ through contacts between H57 and H58 in domain III and H63. The same correlation between the folding of sub-domains IV_5′_ and III_tail_ was observed under bL17 depletion [13], and domain III was shown to fold independently of r-proteins or the rest of the 23S rRNA [37]. Sub-domain III_tail_ is mostly folded in all of the particles, independently of domain IV and uL2, suggesting that it stabilizes domain IV rRNA at the early stages of ribosome biogenesis.

In class 2 particles, the RNA structure is stabilized by additional r-proteins. Unlike in class 1, there is no density for sub-domain IV_5′_ (Figure 7). In class 2-1, the whole flank opposite to the CP (sub-domain III_tail_ and helices in domains 0 and VI) is shifted 20 Å away from the subunit interface compared to the native particle, whereas in class 2-2, this lobe is instead in near-native position.

The exit tunnel is expanded in the smaller particles, in particular in class 2-2 (Figure 3c,d). This would allow access of RlmF to its modification site A1618 in classes 1, 2, and 3-1. These constitute 17% of the total number of particles, in reasonable agreement with 20% of the particles being labeled after in vitro methylation.

Class 3-1 is rather similar to 2-1 but H63 is more pronounced and there is more diffuse density in the H33–H35a region. Correlated to this, domain III_tail_ is closer to its native position, stabilized by H63 and possibly by the higher occupancy of uL3 and bL17. 

In class 3-2, domain II is in a more native position compared to 3-1. It is the smallest particle with unambiguous density for H34–H35 (Figure 7), correlated with improved positioning of nearby domain III (e.g., H58) and H96 and H101 in domain VI.

#### 3.8.2. R-Protein uL3 Is Important for the Stability of Domains 0 and VI

Class 5-1 shares attributes with classes 2-2 and 3-2, but H63 is close-to-natively folded (Figure 7). Similar to in larger class 5 particles, diffuse density extends from H73 across the particle to H22 at lower thresholds (see below).

Higher occupancy of uL3 in 5-1 than in 3-2 seems to cause stronger density for domains 0 and VI (except H96). The same trend is also observed between 2-2 and 2-1. In the mature particle, the 128–153 loop of r-protein uL3 directly interacts with several of these helices (Appendix A), supporting the importance of uL3 in maintaining the native H73 structure, preventing its misfolding (see above).

#### 3.8.3. R-Protein uL2 Stabilizes Sub-Domain IV_5′_

Class 4 has many similarities to 5-1, but the misfolded H73 dislocates a lobe at the solvent side close to the base of the stalk, including H1, H41, uL13, bL20, and bL21. Class 4 also has significantly stronger density for sub-domain IV_5′_ and more mature-like H33–H35a. The differences in domains II and IV seem correlated with increased occupancy of uL2, also seen in classes 5-2, 5-3, 5-5, and 5-6 and in the clustering analysis (Appendix A, respectively).

Class 5-2 is highly similar to class 4 (Figure 4), but with stronger density for sub-domain IV_5′_. 

#### 3.8.4. Helices H79 and H88 Anchor the Core of Domain V

Class 5-3 combines features from 3-2, 5-1, and 5-2 (Figure 6). Class 5-3 has density for r-protein uL19 and low levels for neighboring uL14, similar to the first particle along folding pathway C in [13] (Appendix A).

Class 5-4 is quite similar to class 5-3, but with slightly weaker sub-domain IV_5′_ and uL2. Domain V is on the other hand more well-defined, in particular H79, H88, and the proximal end of H89.

Class 5-5 shows less distinct density for domain V compared to 5-4, although H79 seems ordered. Instead, the density for uL2 and sub-domain IV_5′_ is stronger (even more than in 5-3), placing it in a different path in Figure 6. The distal part of helix H67 is well-defined, but the 3′-half of domain IV is disordered (Figure 7).

#### 3.8.5. Stability of Sub-Domain IV_3′_ Depends on Domain V 

In class 5-6, all the elements that are visible in either of the smaller particles in class 5 are ordered (Figure 6). 

Class 5-7 shows stronger and more well-defined density for domain V than the smaller particles in class 5. There is clear density for helices H74, H75, H76, H79, and H88, albeit shifted from their native positions (Figure 7). Similarly, the folding of domain V along the C/E paths starts with H79 to expand towards the two protuberances under bL17 depletion [13]. The higher occupancy of uL13 in class 5-7 seems to stabilize H1 and H41.

Class 5-8 shows more native structure for domain V, which is partly held in place by sub-domain IV_3′_ (Figure 7), but the distal ends of H68 and H69 are flexible. There is also weak density for the rest of the 3′ end of domain V, including H90–93 and the proximal end of H89. Native positioning of H89, however, requires uL16 [18]. Finally, the H82–H87 branch of domain V, which forms part of the CP, shows partial order, and very weak density can be discerned for additional large CP elements. 

#### 3.8.6. The CP Is Severely Destabilized in the LiCl Core Particles

Class 6 is the only particle where most of the CP rRNA, including 5S rRNA, is folded, but otherwise it is similar to class 5-8. The CP is highly depleted of r-proteins but contains low levels of uL5, uL18, and uL30. Relative to the mature particle, the entire CP has shifted away from the subunit–interface side.

### 3.9. Folding of H49a and H35 Demonstrate Analogies between Disassembly and Assembly

Step-by-step comparison of the smaller to larger particles allows the folding of parts of the LSU to be modeled as an intricate sequential addition of structural layers onto a pre-existing core. As an example, we analyzed the structure of helices around H49a (modification site of RlmF) and H35, in a region of tertiary interactions between domains 0, II, III, IV, and V that is stabilized by the positively charged extensions of r-proteins uL2, uL22, and bL34 (Figure 8 and Appendix A).

#### 3.9.1. Folding and Tertiary Interactions of H49a

Helix H49a is a stem-loop in domain III that bridges to domain II, close to the nascent-chain exit tunnel (Figure 3c,d). In the mature particle, the stem of H49a packs against helices H49b and H50, and the loop forms interactions with H50, H47, and H35 (Figure 8g). Based on this region of the different particles (Appendix A), we reconstructed the sequential assembly of these interactions (Figure 8b–f). In class 2-2 (Figure 8b), H48 and H49b are folded but H49a is completely disordered. In other early classes (2-1 and 3-1), H49a is folded but points away from H50 at different angles. In classes 5-1 (Figure 8c) and 3-2, H49a packs against H49b (contacts 1 and 2 in Figure 8g). Although the loop of H49a lacks most of the native tertiary contacts, there is weak density for A1616 connecting to H50 (contact 3). Class 4 is similar but also shows density for the 5′-end of H47 (contact 4) and H35. In classes 5-3 (Figure 8d) and 5-2, H49a is in a close-to-native conformation. Contact 5 is formed through the stacking between A1618 of H49a and A749 of H35 observed in class 5-4, and the triple-A stack is completed by A1272 of H47 as observed in class 5-5 (Figure 8e). The stack would be further stabilized by the N6 methylation of central A1618 [38]. Classes 5-6 to 5-8 (Figure 8f) show only a slight distortion of the H49a loop at the flipped-out base of A1614 compared to the native structure where it forms an interaction with the extended loop of uL22, with rather weak density for the tip in all classes.

#### 3.9.2. Folding and Tertiary Interactions of H35

Helix H35, a stem-loop in domain II, packs outside H49a, contacting domains 0 (H73), IV (H64 and H65), and V (H93) (Figure 3f and Figure 8g). H35 is disordered in most small classes, such as 2-2 (Figure 8b), and only vaguely discernable in the other small particles. The stability of H35 shows a correlation to the presence of r-protein uL2 (Figure 8c–e). uL2 interacts with H35a, which packs outside of H35 (Figure 8c), and in turn contacts both H35 (contact 6) and H65 (Figure 8d). An A-minor contact interlocks H64 and H35 (contact 7), observed in the more mature-like classes 5-3 and 5-5 (Figure 8d,e), but absent in e.g., class 5-1. The further interactions of H65 with the loop of H35 (contact 8) and its loop to H73 and H93 (contacts 9 and 10) and uL22 are only present in the largest particles, such as class 5-8 (Figure 8f), completing the native structure.

## 4. Discussion

Core particles of the LSU, where the most loosely attached ribosomal proteins and the 5S RNA have been removed using high-salt wash protocols, were first used in studies of ribosome assembly more than half a century ago [2,39]. Here, our results provide a structural understanding of what these particles are, and of their similarities and differences to structures occurring during in vivo and in vitro ribosome assembly. 

The removal of loosely associated r-proteins with a high-salt wash allows access to the smallest stable cores of the large subunit. These particles show density for only 4–8 r-proteins, but additional ones are possibly bound to disordered rRNA regions. With the exception of the smallest particles, the reconstructed cores show strong similarity to intermediates isolated from bL17-deficient cells or from in vitro reconstitution (Appendix A) and their protein content agrees with intermediates formed during in vivo assembly [5]. Similar to results from the purification of in vivo 30S sub-particles at different salt concentrations, “exposure to the high salt buffer does not destroy the known binding interdependences between the ribosomal proteins” [40]. The similarities between states obtained during in vitro disassembly and different variants of assembly strongly support the innate propensity of the rRNA and the r-proteins to adopt distinct, meta-stable complexes. Evolutionary, this inherent property is likely important for the robustness of ribosome assembly upon changing or challenging conditions, such as disabling mutations of an r-protein [13]. Still, local minima in the energy landscape can lead to the trapping of misfolded structures. One such example is the strong non-native secondary structures observed to form by the 3′ strand of helix H73 (Figure 5). This manifests the need for ribosomal proteins not only to facilitate formation of the active RNA structure during co-transcriptional in vivo assembly but also to maintain the native structure in a partly folded structural context. During in vivo assembly, RNA helicases allow remodeling of such misfolded structures [9] and during in vitro assembly, the optimized conditions of temperature and magnesium [1,41] may prevent their formation. Still, also in vivo perturbation by depletion of ribosomal proteins or assembly factors may produce unstable particles that are prone to degradation or misfolding.

The r-protein occupancy of our LiCl core particles largely agrees with previous characterizations [2,6,20,42]. The spontaneously exchangeable r-proteins in vitro and in vivo [7] are the most loosely associated ones, a small subset of the ones that can be dissociated with salt. There is no indication that their dissociation and re-association would cause major structural changes, which seems evolutionary important to maintain the active pool of ribosomes.

The wide range of LSU sub-particles that we observe allows analysis of the consequences of removal of specific r-proteins. For some r-proteins, e.g., uL3, a primary binder, and uL2, a later binder [5] there is a strong link between their presence and observation of the native rRNA structure in their close surrounding. The central protuberance shows high sensitivity to deproteination and unfolds as proteins leave. In contrast, it can fold early during ribosome biogenesis as demonstrated in the bL17-depletion model [13] and during LSU in vitro reconstitution [18]. The structural stability of domain III seems less dependent on r-proteins. 

To what extent does ribosome disassembly in vitro reverse the process of ribosome assembly in vivo or in vitro? Apart from the CP-region proteins uL5, uL15, and uL18, the r-proteins that bind first during assembly are also the ones that stay bound during high-salt wash. The disassembly process is sterically constrained by outer assembly layers of the ribosome, and thereby more similar to in vitro assembly where the structure forms starting from the core and continuing outwards. During the co-transcriptional in vivo process, there is an additional directionality, where at a given time only part of the rRNA can be available for folding and tertiary interactions (reviewed in [9]). However, since functional ribosomes can be assembled on circularly permutated 23S RNA linked to 16S rRNA, no directionality can be required for functional 50S assembly [43]. The 5′ and 3′ ends of 23S RNA together form H99, which remains folded in all our particles (Appendix A), and the ordered elements of the smallest particles are scattered in several blocks along the 23S rRNA sequence, showing no correlation between directionality and stability of RNA structure. In addition, the correlation between the occupancy of r-proteins in our core particles and early binding during in vivo assembly [5] suggests that directionality is not a main factor in r-protein affinity. 

The reconstructed series of particles subjected to r-protein dissociation corroborates the multi-pathway of assembly [13,44,45] by demonstrating the semi-independent stability of folding blocks of rRNA and r-proteins. In particular, since parts of domains IV and V can remain structured while the other unfolds, we can from the LiCl core particles derive two distinct pathways where either of these blocks independently remains folded (Figure 6).

Compared to the well-studied early, co-transcriptional 30S assembly process [46], little is known about the corresponding steps of 50S assembly. However, this study provides examples of concepts that have been described for the 30S, such as the sequential reduction of structural variability of RNA by binding of r-proteins, specifically in regions that require long-range interactions to reach their native structure. There is a clear similarity between how r-proteins during assembly recognize and bind to structured regions and further contribute to the maturation of the nearby structure, and how the dissociation of r-proteins seems to start with the disordering of their extensions and loops coupled to loosening of rRNA structure to be followed by dissociation of the folded r-protein domains and unfolding of the respective rRNA binding sites. The native packing of the RNA structure sometimes requires such r-protein tails or loops, as an example, H49a does not adopt its fully native conformation in absence of the uL22 loop (Figure 8f), the deletion of which has been shown to lead to the accumulation of immature LSU particles [47]. Moreover, the back sides of both subunits show higher stability towards high salt than the interface sides (this study and [40]), in line with preventing the premature association of the subunits. We observe a multi-pathway disassembly process, in agreement with the complex “assembly landscape” of the 30S subunit that was proposed as analogous to the energy landscape in protein folding [48]. These observations support the fact that core particles, produced by high-salt wash, in some respects can be used as mimics of transient early assembly intermediates, for example, in studies of maturation factors. 

In conclusion, the in vitro-generated 50S LiCl core particles represent a collection of intrinsically stable or meta-stable complexes, from which we can learn about the inherent structural properties of ribonucleoprotein particles, and specifically about states that are likely to occur during 50S ribosome assembly. Recent developments in the classification and analysis of heterogeneous particle populations [49,50] will be crucial in further such studies.

## Figures and Tables

**Figure 1 biomolecules-12-01605-f001:**
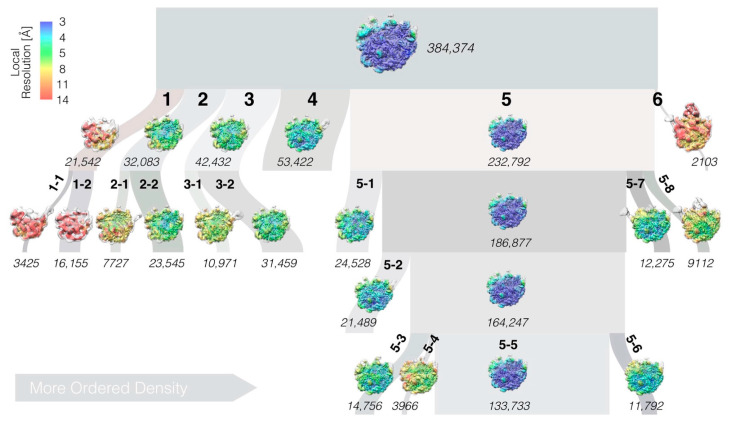
Single-particle reconstruction and hierarchical classification of LiCl core particles. The widths of the branches in the diagram reflect the number of particle images (italic font) in each class (bold font). Classes are sorted from left to right, approximately according to the amount of ordered density. Images show the reconstructions in “crown view” with the L1 protuberance to the left and the L7/L12 stalk to the right (cf. Appendix A for additional views).

**Figure 2 biomolecules-12-01605-f002:**
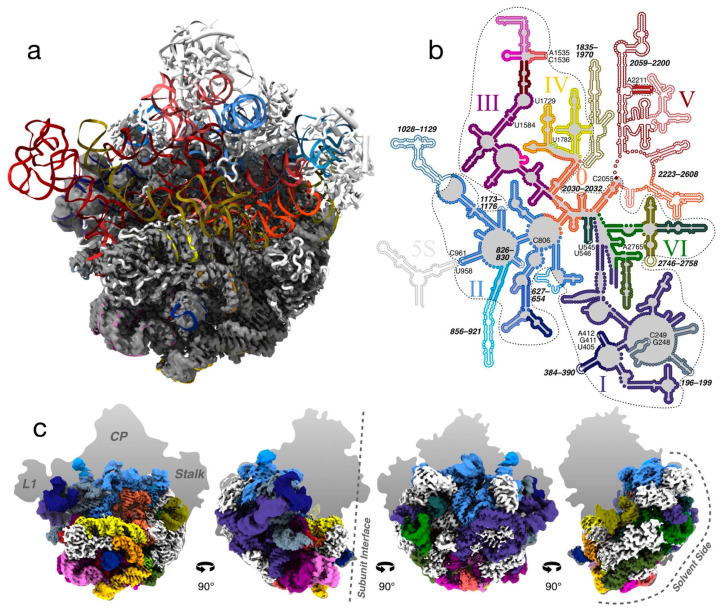
Consensus reconstruction of the 50S LiCl core particle. (**a**) The map (gray) in “crown view”, with a rigid-body docked model of the mature 50S particle (PDB ID 4YBB) colored as in (**b**). (**b**) Secondary-structure map of the LSU rRNA (adapted with permission (CC BY-SA 3.0) from Ref. [34]). Folded parts in the consensus particle are indicated with filled circles and outlined with a dashed line. Disordered helices, loops, or single nucleotides are shown as white dots and labeled. The RNA helices are colored in orange, indigo, blue, purple, yellow, or red tones for domains 0–VI, respectively. (**c**) Four different views of the map with the rRNA colored according to (**b**) and r-proteins in white.

**Figure 3 biomolecules-12-01605-f003:**
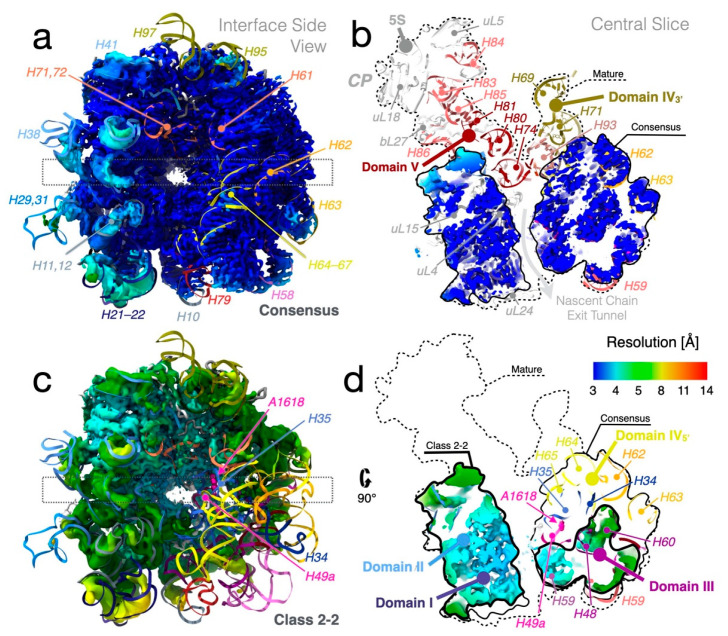
Local resolution and the expanded exit tunnel. (**a**) Consensus reconstruction seen from the interface side, centered on the nascent-chain exit tunnel, colored according to local resolution, with the refined atomic model colored according to Figure 2b. (**b**) Central slice through the map, rectangle in (**a**), showing the expanded exit tunnel and the absence of the CP. The rigid-body fitted model of the mature 50S particle (PDB ID 4YBB [28]) is colored as in Figure 2b. The solid black line indicates the extent of the ordered density and the dashed line indicates the extent of the mature particle. (**c**,**d**) Class 2-2 in the same view and model as in (**a**,**b**). In this particle, sub-domain IV_5′_ and parts of domains II and III are disordered compared to the consensus structure, in particular, nucleotide A1618 in H49a. The thick black line in (**d**) indicates the extent of the density in class 2-2, while the thin and dashed lines are the same as in (**b**).

**Figure 4 biomolecules-12-01605-f004:**
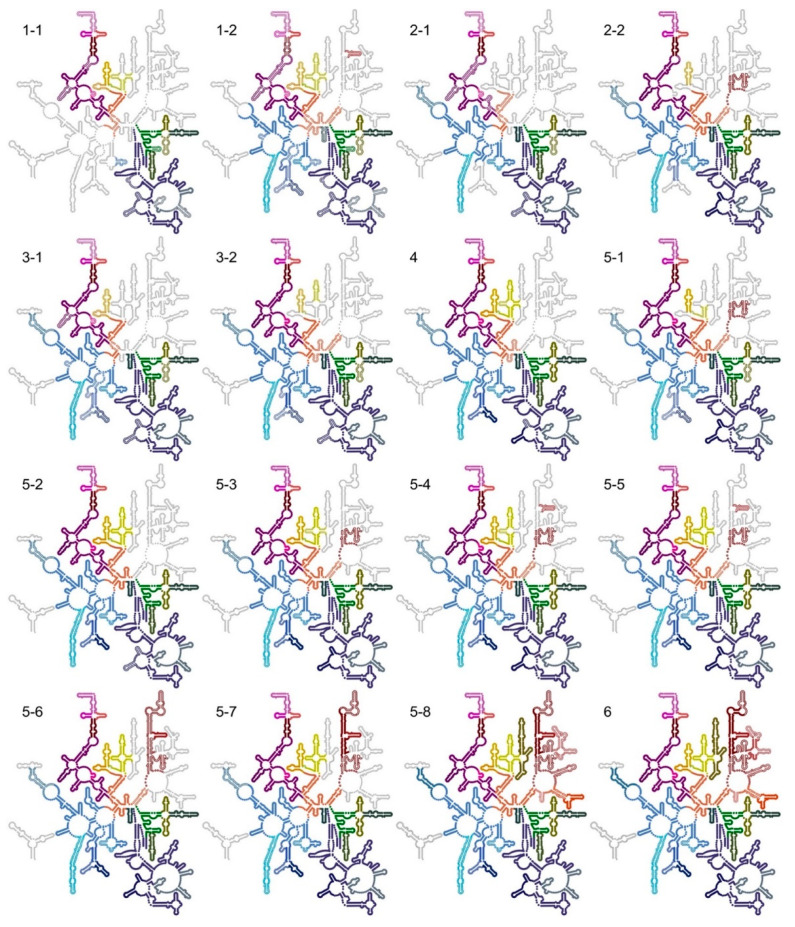
Ordered rRNA for all particle classes mapped on the secondary-structure map of the LSU (adapted with permission (CC BY-SA 3.0) from Ref. [34]). Ordered RNA helices are colored according to Figure 2b. Partially ordered helices have half-filled circles (lighter color) and disordered helices are gray.

**Figure 5 biomolecules-12-01605-f005:**
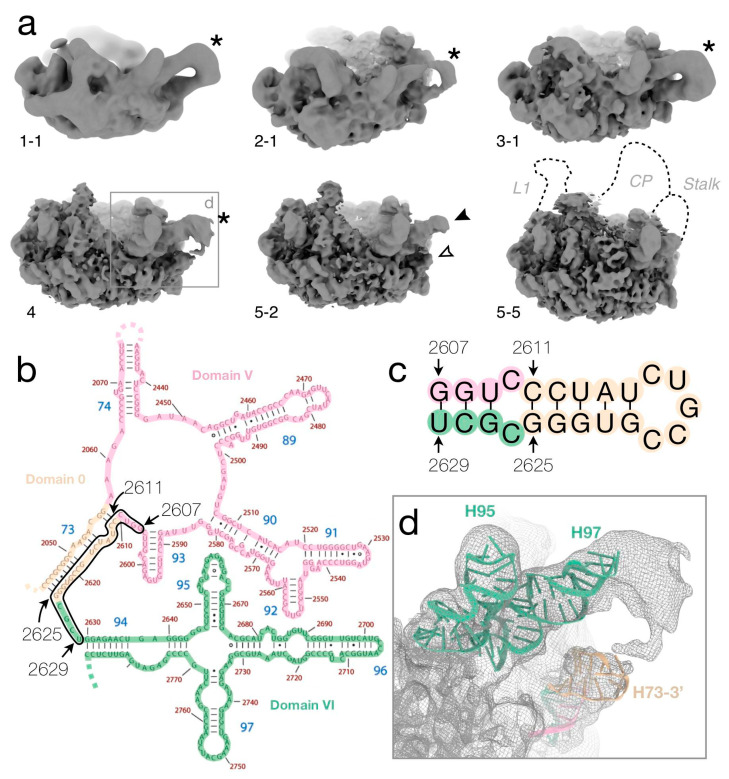
Non-native density close to the base of the L7/L12 stalk. (**a**) In classes 1-1, 2-1, 3-1, and 4, the density of H97 extends and loops back (asterisk) at lower map thresholds. In class 5-2, the density for the upper (black arrowhead) and lower (white arrowhead) arms are not connected. Analogous classes 1-2, 2-2, 3-2, 5-1, and 5-3, respectively, do not show this extra density (not shown). (**b**) Secondary structure map of the *E. coli* 23S rRNA shown for parts of domains 0, V, and VI (adapted with permission (CC BY-SA 3.0) from Ref. [34]). The black line encloses the segment that was used for the secondary-structure prediction in (**c**). (**c**) Predicted secondary structure of the 3′-strand of H73 and neighboring nucleotides. Colors as in (**b**). (**d**) Loop density for class 4 (rectangle in (**a**)) with a rigid-body docked model of the 8-bp stem-loop in (**c**) predicted by 3dRNA [35] colored as in (**c**). Models of H97 and H95 are shown for reference. The density connecting H97 and H73-3′ could represent an unidentified protein.

**Figure 6 biomolecules-12-01605-f006:**
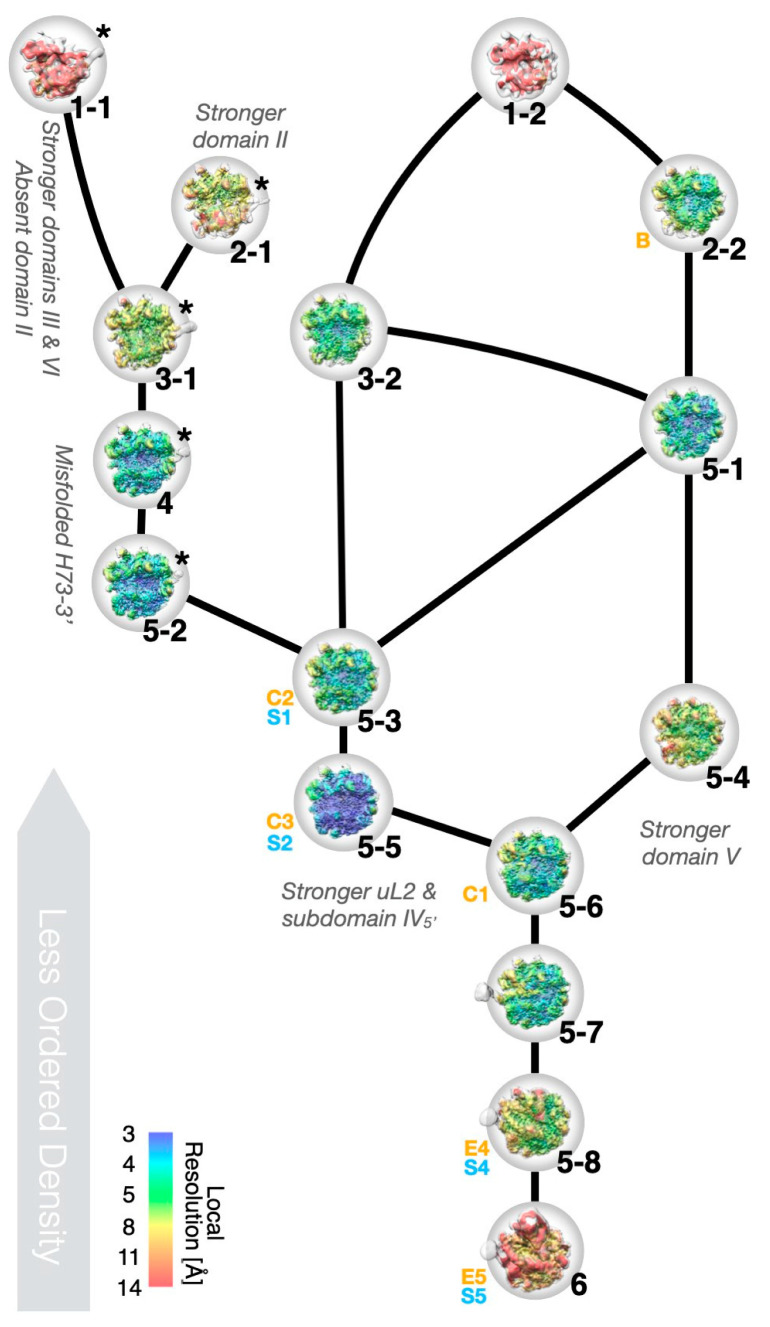
Relations between particle classes. Similar particles are connected by lines, representing possible disassembly paths during salt wash. The right side contains particles with stronger density for domain V, while the left-side particles have stronger densities for uL2, H34–35, and sub-domain IV_5′_. The left side-shoot consists of particles with misfolded H73 (asterisks, Figure 5). The particles are shown as in Appendix A. Orange and blue designations are similar classes in Davis et al. [13] and Nikolay et al. [18] (see also Appendix A).

**Figure 7 biomolecules-12-01605-f007:**
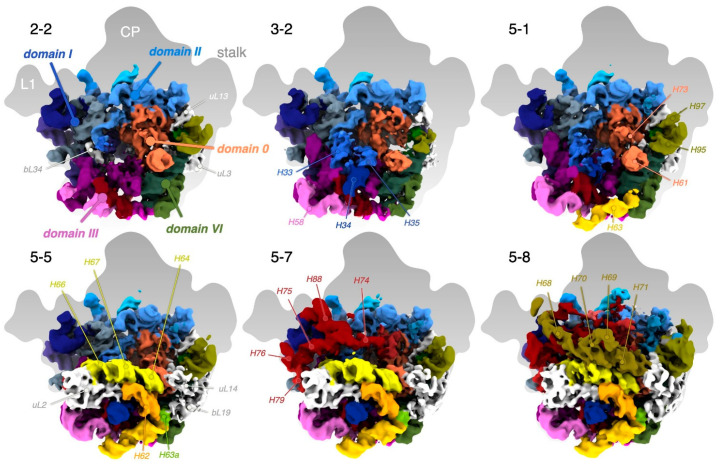
Side-by-side comparisons of particles of different sizes. RNA is colored as in Figure 2b and proteins are shown in white.

**Figure 8 biomolecules-12-01605-f008:**
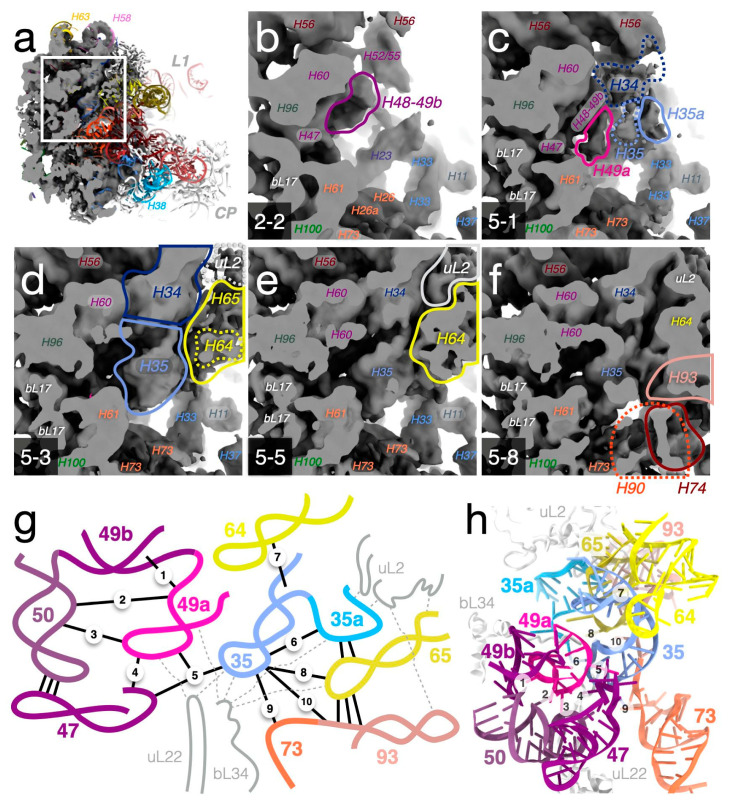
Structure of the H49a-H35 region in different particles suggests a local order of assembly. (**a–f**) Maps showing varying degrees of folded structure in this region. Lines indicate densities that differ between maps (the dashed lines indicate weak/diffuse densities). The maps are not sharpened, or low-pass filtered, and are shown at a level of 0.05. Appendix A shows the same view for all particle classes. (**a**) The overview shows the consensus map and a mature model, PDB ID 4YBB, colored as in Figure 2b. (**b**) In class 2-2, H48 is folded, but not H49a. (**c**) Densities for H34, H35, and H35a outside of H49a in class 5-1. (**d**,**e**) Occupancy for H64, H65, and uL2 is partial in class 5-3, but close to full in 5-5. (**f**) Helices H90, H93, and H74 are folded into class 5-8. (**g**) Schematic view of stabilizing interactions in the H49a-H35 region. Tertiary contacts for H49a and H35 are indicated with solid lines and numbered 1–10 in the approximate assembly order, as deduced from the maps (see Section 3.9.1 and Section 3.9.2). Dashed lines indicate contacts to r-proteins. (**h**) Structure of helices and r-proteins in (**g**) in the mature 50S (PDB ID 4YBB), tertiary contacts are numbered 1-10 as in (**g**).

## Data Availability

Atomic coordinates for the consensus structure have been deposited in the Protein Data Bank under accession code 7ODE. All maps have been deposited in the Electron Microscopy Data Bank with accession codes EMD-12826 for the consensus reconstruction and EMD-12828 through EMD-12841, EMD-12843, and EMD-12844 for the different classes.

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
