# Peer review of "Structural Consequences of Deproteinating the 50S Ribosome"

_biomolecules, 2022, doi:10.3390/biom12111605_

Round 1

Reviewer 1 Report

Biomolecules – Structure of the 50S after deproteinization

Ribosomes are complex aggregates of RNA and proteins, essential for protein synthesis in all biological cells. The assembly of these complex structures from their constituents has been under investigation since the 1970s. By now, we understand the principles, but not all the details. One outstanding question is how subdomains of the ribosomal subunits are stabilized and to what extent the stability of these subdomains depends on each other. Larsson, Kanchugal, and Selmer have submitted a manuscript determining the structure of “core particles” remaining after removing a subset of ribosomal proteins from E. coli 50S ribosomal subunits with 3.5 M LiCl. Careful experiments using LC-MS/MS chromatography and cryo-electron microscopy determined the structure which lacks several of the well-known subdomains of the native 50S. The authors conclude that ribosomal proteins stabilize rRNA structures and that the core particles share most, but not all, features with assembly intermediates from both in vivo biogenesis and in vitro reconstitution. Although none of these conclusions are surprising, they add to our understanding of the cohesiveness of ribosomes. The presentation is well-written and contains excellent figures.

Specific comments:

·      The degree of rRNA structural integrity in the core particles should be documented.

·      Please use original references rather than reviews when describing the salient features of ribosome biogenesis and the core particle phenomenon. It seems especially relevant to refer to the work of Littlechild and her coworkers on the stepwise LiCl stripping of ribosomal proteins.

·      It is relevant to point out that particles accumulating in ribosomal protein mutants or after depletion of specific ribosomal proteins or assembly factors may also be degradation intermediates of unstable particles caught in kinetic traps.

Author Response

Answers to comments from referee 1.

The degree of rRNA structural integrity in the core particles should be documented.

Answer: Thanks for the suggestion. We agree that this information is not provided in an easily digestible format. This data was present in the heat maps presented in figure S12, but we have now added an additional main Figure 4, showing the ordered RNA of the different particles mapped on the secondary structure of 23S RNA. We believe that this makes the rRNA structural integrity in the different particles much clearer.

Please use original references rather than reviews when describing the salient features of ribosome biogenesis and the core particle phenomenon. It seems especially relevant to refer to the work of Littlechild and her coworkers on the stepwise LiCl stripping of ribosomal proteins.

Answer: Thanks for pointing this out. In the introduction, we have changed the references regarding the first demonstration of LiCl core particles to Homann & Nierhaus European Journal of Biochemistry 1971 and added references to the work of Littlechild regarding the stepwise LiCl-mediated stripping of ribosomal proteins in non-denatured state (Morrison et al., FEBS Lett 1977, Dijk & Littlechild Meth Enz 1979).

It is relevant to point out that particles accumulating in ribosomal protein mutants or after depletion of specific ribosomal proteins or assembly factors may also be degradation intermediates of unstable particles caught in kinetic traps.

Answer: Thanks for pointing this out. We have attempted to clarify that these strategies are indeed perturbations of the native ribosome assembly. We re-wrote the sentence on lines 62-65 to the following: “To enrich ribosome assembly intermediates in vivo for these studies, bacteria have been perturbed by assembly inhibitors [11,12], knock-down of r-proteins or assembly factors [13–15] or subjected to pull-downs with assembly factors as bait [16,17].”, and added a sentence to the discussion (line 604-606): “Still, also in vivo perturbation by depletion of ribosomal proteins or assembly factors may produce unstable particles that are prone to degradation or misfolding.”

Reviewer 2 Report

This is a very interesting paper that revisits old experiments in which ribosomal subunit "cores" were created by stripping outer/loosely bound ribosomal proteins by means of LiCl washing. Although it was known that such cores had physiological significance, as they could be reconstituted to active particles by adding the missing proteins, the matter remained more or less unexplored. Here the authors apply cryo-electron microscopy to sort out the various types of "cores" obtained by LiCl washing, and to investigate their structure in detail. In this way, and by comparing their results with extant literature data in vivo and in vitro, they reach interesting conclusions about the pathways of ribosome assembly and the role of ribosomal proteins in stabilizing rRNA structure. It is certainly worth publishing.

Author Response

Answer: Many thanks for positive comments.

Reviewer 3 Report

The manuscript by Larsson at all represents work focused on deciphering the assembling pathway of the 50S ribosomal subunit. It should be stated, that maturation of mega-Dalton molecules, such as ribosomal subunits is composed of highly complex way of events, where cis-acting elements and plethora of trans-acting factors are involved in assisting rRNA to be correctly folded into translationally competent ribosome. The maturation pathways for bacterial and eukaryotic ribosomes were extensively studied, with prominent example of the 30S in vitro assembling into active form, few decades ego. Recently, as Authors acknowledge in their manuscript, the cryo-EM approach provided enormous insight into understanding of the ribosome maturation, including number of elegant work presenting the pre-60 and pre-40 assembling pathways; also, the bacterial ribosomes did not escape attention in connection with the maturation, especially the 50S subunit maturation was characterized with the aid of cryo-EM approach, laying the foundation for understanding of the assembling pathway of this bacterial ribosomal subunit. For example, in the previous elegant publication, from Christian Spahn group, authors used in vitro reconstitution of the purified ribosomal RNA and protein components under defined and scalable conditions, and solved the intermediates of 50S at resolutions of 4.3–3.8 A; additionally, James Williamson research group has provided profound insights into the 50S maturation, especially recent publication (Structure, 2022, 30(4), pp. 498–509.e4) is showing the developed workflow applied to a dataset of E. coli 50S ribosome assembly intermediates.

Authors in the presented manuscript made an attempt to characterize the assembling pathway of the 50S ribosomal subunit. They used the salt-washed core 50S particles for the cryo-EM analysis and by sorting approach they were able to identify 6 major types of 50S sub-particles. Authors applied so called top-down analysis, building the view about 50S maturation based on the disassembled 50S particle, but it should be stressed that information that was obtained does not take us beyond the current understanding. First of all, the main obstacle: they used extremely harsh conditions for 50S disassembling (3.5 M LiCl for 5 hours), and such conditions are causing significant collapse of the 50S in terms of structure and released proteins (it looks like majority of r-proteins are released, and there is no significant differences between sub-populations, as presented in MS data); it is mandatory to use gradient of LiCl concentration, to obtain several intermediates, based on homogeneous well sorted 50S populations, which might resemble biologically significant assembling sub-populations. A consequence of authors approach is low resolutions structures based in low number od suitable particles; it looks like only population #5-5 has satisfactory resolution, in the range of 3-4A, and the rest of provided models exceed 4A and more, making interpretation of the data very difficult or almost impossible (difficult to build reliable data based on 2000 or so). Therefore, to exclude the possibility of artefactual observations, authors should consider significant extension of the experimental approach, not only rely on disassembling approach (usage of the LiCl different concentration) but they should consider isolation of natively formed intermediates. Thus to confront top-down with bottom-up analysis will bring biologically significant data. At current stage, presented data should be considered as a preliminary approach to start; the idea is interesting, the technology is in the hands of authors, however the data represent border-line artefactual observations, thus, the provided information may lead astray.

Author Response

Answers to comments by referee 3:

The manuscript by Larsson at all represents work focused on deciphering the assembling pathway of the 50S ribosomal subunit.

Answer: This is a misunderstanding of the aim of the present study. The title, abstract and introduction clearly states that the focus of the work is not to decipher the assembly pathway, but instead to investigate the nature of ribosomal sub-particles formed during salt-induced disassembly of the 50S subunit. We have re-written the introduction slightly (moving down the introduction to in vivo biogenesis) to make this even more clear. See for example title: “Structural consequences of deproteinating the 50S ribosome”, lines 7-10: “Ribosomes are complex ribonucleoprotein particles. Purified 50S ribosomes subjected to high-salt wash, removing a subset of ribosomal proteins (r-proteins), were early shown competent for in vitro assembly into functional 50S subunits. We here used cryo-EM to determine the structure of such LiCl core particles derived from E. coli50S subunits.” and lines 83-85: “We here set out to further characterize the structure of LiCl core particles using single-particle cryo-EM, with the goal to at high-resolution characterize the structure and composition of these particles.”

It should be stated, that maturation of mega-Dalton molecules, such as ribosomal subunits is composed of highly complex way of events, where cis-acting elements and plethora of trans-acting factors are involved in assisting rRNA to be correctly folded into translationally competent ribosome. The maturation pathways for bacterial and eukaryotic ribosomes were extensively studied, with prominent example of the 30S in vitro assembling into active form, few decades ego.

Answer: This is clearly stated in the introduction and also referred to in the discussion. See for example lines 42-62: “In vivo, E. coli makes use of a multitude of auxiliary RNA modification enzymes and ribosome assembly factors to fold and assemble ribosomes from three rRNAs and 54 r-proteins (reviewed in [8,9]). Despite the complexity of the process, the assembly time for a mature ribosome in vivo in exponentially dividing E. coli has been estimated to only 2 minutes [10]. Our structural understanding of folding and assembly of ribosomes has greatly benefitted from progress in cryogenic electron microscopy (cryo-EM), where classification schemes in single-particle reconstruction have proven essential to study these heterogeneous particle ensembles.”

However, since the focus of the current work is to investigate salt-induced disassembly and compare the identified structures to states formed during ribosome assembly, we do not go through the entire assembly literature, and mainly focus on the 50S. We do refer to 30S assembly in the discussion (lines 645-649) and refer to important work by Sarah Woodson.

Recently, as Authors acknowledge in their manuscript, the cryo-EM approach provided enormous insight into understanding of the ribosome maturation, including number of elegant work presenting the pre-60 and pre-40 assembling pathways; also, the bacterial ribosomes did not escape attention in connection with the maturation, especially the 50S subunit maturation was characterized with the aid of cryo-EM approach, laying the foundation for understanding of the assembling pathway of this bacterial ribosomal subunit. For example, in the previous elegant publication, from Christian Spahn group, authors used in vitro reconstitution of the purified ribosomal RNA and protein components under defined and scalable conditions, and solved the intermediates of 50S at resolutions of 4.3–3.8 A; additionally, James Williamson research group has provided profound insights into the 50S maturation, especially recent publication (Structure, 2022, 30(4), pp. 498–509.e4) is showing the developed workflow applied to a dataset of E. coli 50S ribosome assembly intermediates.

Answer: We fully agree that the Williamson and Spahn groups have made important contributions to the current understanding of 50S ribosome assembly. However, the present study as explained above has a different aim. Yet, extensive parallels are observed between the complexes identified upon salt-induced removal of r-proteins and assembly intermediates generated in vitro and in vivo. We think is very noteworthy, supporting what we write in the abstract: “Many of these particles showed high similarity to in vivo and in vitro assembly intermediates, supporting the inherent stability or metastability of these states.”. The paper by Rabuck-Gibbons et al., Structure, 2022, from the Williamson group, is a follow-up study re-analyzing the bL17 depletion data set (Davis et al. 2016). We have added this reference (line 668-669): “Recent developments in classification and analysis of heterogenous particle populations[49,50] will be crucial in further such studies.”

Authors in the presented manuscript made an attempt to characterize the assembling pathway of the 50S ribosomal subunit.

Answer: See above, the aim of this study was not to characterize the assembly pathway of the 50S subunit.

They used the salt-washed core 50S particles for the cryo-EM analysis and by sorting approach they were able to identify 6 major types of 50S sub-particles. Authors applied so called top-down analysis, building the view about 50S maturation based on the disassembled 50S particle, but it should be stressed that information that was obtained does not take us beyond the current understanding. First of all, the main obstacle: they used extremely harsh conditions for 50S disassembling (3.5 M LiCl for 5 hours), and such conditions are causing significant collapse of the 50S in terms of structure and released proteins (it looks like majority of r-proteins are released, and there is no significant differences between sub-populations, as presented in MS data); it is mandatory to use gradient of LiCl concentration, to obtain several intermediates, based on homogeneous well sorted 50S populations, which might resemble biologically significant assembling sub-populations. 

Answer: We fully agree that a salt gradient would have been a better approach if our aim was to obtain states more similar to late ribosomal assembly intermediates. However, this was not our aim. A starting point for this work was the finding that LiCl core particles after wash with 3.5 M salt worked as in vitro substrate of  rRNA methyltransferase RlmF (Sergiev et al., JMB 2008). That prompted our interest in investigating in what types of subribosomal particles the modification site A1618 would be accessible for modification. These particles would need to be less assembled than intermediates from in vitro and in vivo assembly studies, thus we selected “harsh” conditions for disassembly, dissociated most proteins and discovered smaller core particles than previously discovered. The modification site was indeed accessible in some of the small particles (classes 1, 2 and 3-1), and we could decipher the different layers of assembly in this region of intricate tertiary interactions (Fig. 8).  We also identified the smallest ribosomal sub-particles to date, where domain II is completely disordered. We are very careful in how we express our findings in relation to studies of ribosome assembly, to not overstate the significance of out work for the in vivo situation; e.g lines 662-664: “These observations support that core particles after salt wash in some respects can be used as mimics of transient early assembly intermediates, for example in studies of maturation factors.”

In addition, the method of producing LiCl core particles has previously been extensively studied and found to be gentle compared to e.g. acid urea treatment (Morrison et al., FEBS Letters 1977).

Regarding the MS data, there was no attempt to check the protein content of different sub-populations. This was only done to monitor r-protein dissociation in the mixed population of LiCl core particles that was applied to the cryo-EM grid (see line 156-159: The pellets were resuspended in 10 uL of imaging buffer and stored on ice before being flash frozen on cryo-EM grids. The protein content was verified by SDS-PAGE and LC-MS/MS analysis of the lane cut from the SDS-PAGE gel.”).

A consequence of authors approach is low resolutions structures based in low number of suitable particles; it looks like only population #5-5 has satisfactory resolution, in the range of 3-4A, and the rest of provided models exceed 4A and more, making interpretation of the data very difficult or almost impossible (difficult to build reliable data based on 2000 or so).

Answer: The limited resolution of some of the particles is exactly the reason that we only built and refined one atomic model, and to a large extent analyzed similarities and differences based on map comparison that show clear features even at low resolution (e.g. Figures 5, 7 and 8). Since we mainly focus on presence/absence of low-resolution elements, the resolutions of the complexes are sufficient to support the conclusions drawn in the manuscript. Most classes have similar resolution as in the studies of assembly intermediates that use for comparisons (Table S2). Seven out of 16 classes are resolved at better than 3.8 Å (equal to the highest-resolution structure in Nikolay et al. 2018). Five classes show 4–6.5 Å resolution (the resolution range of the 14 classes in Davis et al. 2016). One class is resolved to 7.6 Å resolution, a resolution good enough to resolve alpha-helices. The remaining three classes (1-1, 1-2 and 6) show 9–10 Å resolution, sufficient to annotate globular parts of most r-proteins and RNA helices (Rabuck-Gibbons et al., Structure, 2022). We have added a clarifying statement on line 303-304: “All particles have resolutions that allow interpretation on the level of presence or absence of double-helical RNA elements”. We do not go into atomic-level description of these classes and explicitly write when the density is not clear enough to make firm conclusions.

Therefore, to exclude the possibility of artefactual observations, authors should consider significant extension of the experimental approach, not only rely on disassembling approach (usage of the LiCl different concentration) but they should consider isolation of natively formed intermediates. Thus to confront top-down with bottom-up analysis will bring biologically significant data. At current stage, presented data should be considered as a preliminary approach to start; the idea is interesting, the technology is in the hands of authors, however the data represent border-line artefactual observations, thus, the provided information may lead astray.

Answer: These suggestions outline a different experimental study with a different aim, but is nothing that we can do within the 10 days of manuscript revision. We do not agree that our observations are artefactual, see above for explanations regarding the careful interpretation of cryo-EM maps at different resolutions. In the manuscript, we present several findings contributing to our understanding of ribosomal stability and disassembly. As examples, analysis of the consequences of removal of specific r-proteins demonstrates impacts of the interactions of r-proteins with rRNA, e.g. how uL2 and uL3 stabilize the native rRNA structure in their surroundings, and that helix H73 can mis-fold upon de-proteination.

Round 2

Reviewer 3 Report

All explanations are sufficient.